# Allergies, Allergic Comorbidities and the Home Environment in Pediatric Asthma in Southern Florida

**DOI:** 10.3390/ijerph18084142

**Published:** 2021-04-14

**Authors:** Nadia T. Saif, Gary I. Kleiner, Lourdes Q. Forster, Eugene R. Hershorin, Andrew A. Colin, Mehdi Mirsaeidi, Naresh Kumar

**Affiliations:** 1Department of Public Health Sciences, University of Miami Miller School of Medicine, Miami, FL 33136, USA; nadia.thura@gmail.com; 2Department of Pediatrics, University of Miami Health System, Miami, FL 33136, USA; gary.kleiner@med.miami.edu (G.I.K.); lforster@med.miami.edu (L.Q.F.); ehershorin@med.miami.edu (E.R.H.); 3Division of Pediatric Pulmonology, Department of Pediatrics, University of Miami Miller School of Medicine, Miami, FL 33136, USA; acolin@med.miami.edu; 4Pulmonary, Allergy, Critical Care and Sleep Medicine Division, University of Miami Miller School of Medicine, Miami, FL 33136, USA; msm249@med.miami.edu

**Keywords:** pediatric asthma, environmental allergy, allergic rhinitis

## Abstract

*Background:* Environmental exposure is critical in sensitization to environmental allergens and pediatric asthma morbidity, especially in tropical climates where children are perennially exposed to bioaerosols, such as pollen and mold spores, and endotoxins. *Objective*: This cross-sectional study examines the association of allergies, associated allergic comorbidities, and the home environment separately and synergistically in pediatric asthma, including in asthma prevalence, severity of asthma, and undiagnosed asthma, in South Florida. *Methods:* An online survey was administered to the parents of children attending two of the University of Miami pediatric clinics from June to October 2016. Descriptive, factor, and multivariate regression analyses were used to analyze the data. *Results*: Of 163 children, 22% (36) children had physician-diagnosed asthma; 10% and 32% had allergic rhinitis diagnosis and rhinitis symptoms, respectively, in the past. The allergy diagnosis age was 2.3 years higher than the asthma diagnosis age (*p* < 0.01). Children with ≥ 2 allergies were 12.8 times more likely to have physician-diagnosed asthma than those without allergies (*p* < 0.01). Children with allergies and allergic rhinitis were 4.3 (*p* < 0.05) times more likely to have asthma, and those with asthma were 15 (*p* < 0.05) times more likely to have an asthma attack than those without known allergies and allergic rhinitis. *Conclusion*: Allergies and associated comorbidities are risk factors of asthma, asthma persistence, and multiple allergies exacerbate their effects. Early screening for allergies and treatment are warranted to manage asthma. Since the home environment plays an important role in sensitization to allergens, further research is needed to assess home-environment-mediated allergic conditions in the onset and persistence of asthma.

## 1. Introduction

Pediatric asthma remains one of the most common chronic diseases in the US, affecting 6.2 million children and accounting for ~$50 billion/year in healthcare costs [1]. Emerging research suggests that the classification of the disease based on etiology and pathophysiological mechanisms, and clearly defining the link between asthma and allergy, is more complex than once thought [2]. Simplistically, asthma is classified into atopic (induced by external triggers, including allergies) and non-atopic. Although mortality due to asthma has declined, there has been an increase in asthma morbidity, especially due to atopic asthma [3], which accounts for > 75% of asthma cases, is the most important phenotype in childhood, and often persists into adulthood [2,4]. The International Study of Asthma and Allergies in Childhood (ISAAC) demonstrates a global trend towards increasing prevalence of asthma and related allergic diseases, namely rhinoconjunctivitis and eczema, particularly among younger children [4]. Therefore, it is important to understand differences in the age of diagnosis of asthma, allergies, and associated comorbidities. External triggers play a vital role in allergic asthma, not only because they impact the onset, progression, and persistence of asthma but also because they increase the risk of both allergic comorbidities, including allergenic rhinitis. Although the role of allergy in asthma has been subject to research scrutiny [5], it sheds little insight into the relative contributions of different allergens separately and synergistically and home environment in pediatric asthma. In South Florida specifically, given the tropical climate, children are perennially exposed to elevated levels of bioaerosols (i.e., airborne microorganism, mold, and pollen spores and endotoxins, hereinafter) which can increase the risk of allergic sensitization [6,7]. This paper aims to address this research gap by examining the age of diagnosis of asthma and allergies, and separate and synergistic effects of different allergies and associated allergic comorbidities on pediatric asthma, asthma severity and wheezing (a crude proxy of undiagnosed asthma, hereinafter) in South Florida.

## 2. Methods and Material

A cross-sectional design was used for this study. Data were collected at two general pediatrics clinics at the University of Miami in Miami, Florida, between June to October 2016. Parents and guardians of children being seen at these clinics were consented. A bi-lingual (English and Spanish) survey was designed in Qualtrics. For parents/guardians of children, who were willing to complete the survey while in the waiting room, an iPad with the consent form was given to them. Those consented to participate were directed to the survey page in the language they chose. The survey included questions about demographics, history diagnosis asthma, symptoms of asthma and other allergic disease, medication use, and home environment, such as exposure to secondhand smoke, pets, and other indoor allergens. Questions were drawn from NHANES 2005–2006 asthma and allergy content questions, BRFSS Asthma Call-Back Survey 2014 Child Questionnaire, and the International Study of Asthma and Allergies in Childhood: Phase Three Core Questionnaire (see Appendix A) [8,9,10]. This study was approved by the University of Miami Institutional Review Board (IRB # 20160040).

Outcomes of interest included asthma diagnosis, undiagnosed asthma, and asthma severity. Allergic comorbidities included allergic rhinitis, eczema, non-infectious rhinitis symptoms (sneezing, runny, or blocked nose in the absence of cold or flu), and testing positive for any allergen based on environmental allergy testing in the past. Asthma diagnosis, allergic rhinitis, and eczema were based on parents’ self-report of a physician diagnosis of these conditions at any point in the child’s lifetime. Wheezing history (ever had wheezing or whistling in the chest; wheezing in the past 12 months) was asked only of parents who reported their child did not have a history of asthma diagnosis. Therefore, wheezing history served as a “crude” proxy of potentially undiagnosed asthma, which is shown to have 75% sensitivity and 64% specificity as a standalone asthma screening question [10]. Outcomes that served as indicators of asthma severity were asked only of children with a history of asthma diagnosis. This included asthma exacerbation (attack) in the past year, and a composite variable that was scored as positive if a “yes” response was recorded for at least 2 of the 4 following outcomes: missed school in the past year, asthma symptoms within the past 30 days, asthma attack in the past year, and asthma medication use in the past 30 days.

Descriptive statistics, factor analysis, univariate and multivariate regression analyses were employed to analyze data. Factor analysis is a data reduction technique that groups correlated variables into unique factors. The main usage of the method is to compute unique factors (i.e., composite scores of variables that have high factor loadings in a given factor. Thus, it addressed the issue of correlation among casual variables. It can also be used to identify correlated (or similar) variables. Thus, to examine similarities in allergic comorbidities, factor, analysis was conducted and factor scores of allergic comorbidities and allergies were computed to assess collinearity among them. Risks of asthma, asthma severity indicators, and wheezing in the past year among non-asthmatic children (potentially undiagnosed asthma) were examined with respect to allergic comorbidities, home environment, and environmental allergens using multivariate logistic regression. In the univariate analysis, each variable was examined separately. In the multivariate analysis, all odds ratio were adjusted for age and gender and computed with robust standard error in STATA [11].

## 3. Results

Table 1 shows demographic, clinical, and home environment characteristics of the study participants. Of the 233 parents/guardians we contacted, a total of 163 (69.9%) participated in the study. Of these 163, 36 (22%) reported that their children have physician-diagnosed asthma; 10% and 32% have allergic rhinitis and rhinitis symptoms, respectively; 18 (11%) were tested positive for environmental allergies in the past (Table 1). Of those tested positive for environmental allergies, the most commonly reported were pollen, followed by dust mite and pet dander. Among children without an asthma diagnosis, 39 (23.9% of the study participants) reported that their child had wheezing in the past, and of these, 12 reported wheezing in the past year. Nearly half (48.5%) of the parents interviewed reported pets in the home, and 4.8% of parents reported a smoker in the home. The majority of homes kept temperatures in the 71–75 °F. 33.1% of participants had an air cleaner or purifier in their home. 42.9% of participants reported seeing a cockroach and 4.8% smelled mold in the past month. Slightly over one-quarter of parents reported carpet in the child’s bedroom (28.5%). 42% of participants pay attention to outdoor pollen sometimes, most of the time, or always (42%), but only 14.8% accessed online pollen or allergy information from sites like allergy.com, weather.com, or other local, state, or private agencies.

Figure 1 shows the age of diagnosis of asthma and allergic comorbidities. The mean age of eczema diagnosis was lowest at 2.45 years, followed by wheezing, asthma, rhinitis symptoms, and allergic rhinitis diagnosis. Testing for allergies (including environmental allergies) occurred at a mean age of 5 years, significantly higher than the age of asthma diagnosis (2.34 year versus 5.02 years; difference = 2.3 years *p* < 0.01; Table 2). Table 3 shows the odds of asthma, asthma severity, and wheezing by the results from past environmental allergen testing. Among all allergens, odds of asthma prevalence was highest for pet dander, followed by allergies for mold, cockroach, and pollens. The odds of asthma was 6 times higher for children who were tested positive for one allergy (odds ratio (OR) 6.39; 95% confidence interval (CI): 1.3–31.8; *p* < 0.01), which doubled for children with ≥ 2 allergies (OR: 12.78; CI: 2.9–57.2; *p* < 0.01).

All allergic comorbidities showed a significant association with asthma risk. Allergic rhinitis had the most significant association with asthma (OR: 10.13; CI: 4.17 to 24.58; *p* < 0.01) (Table 4). For asthma severity score ≥ 2, several variables demonstrated protective odds ratios, including age (OR: 0.73; CI 0.58–0.91; *p* < 0.01), having pets in the home (OR: 0.07; CI 0.01–0.58; *p* < 0.05), and washing bed sheets in hot or warm water compared with cold water. For asthma attack, the odds ratio for rhinitis symptoms and asthma attack could not be estimated due to the small sample size, but the relationship was but significant by chi-square association; all 13/13 children who had an asthma attack in the past year reported rhinitis symptoms, compared with 13/21 (61.9%) who did not have an asthma attack, *p* < 0.05. Similarly, the odds ratio for accessing pollen or allergy info in the past week and asthma attack could not be estimated, but all who accessed pollen/allergy information did not report an asthma attack (*p* = 0.08).

Factor analysis confirms that allergic comorbidities and allergies are highly correlated. For example, rhinitis, running nose, environmental and other allergies, and hay fever have high positive loadings in Factor 1, which accounted for 68% of the total variance (Table 5). Both asthma prevalence and asthma attack were included in the factor analysis in order to understand their collinearity (associations) with common allergic comorbidities (Table 5). History of environmental allergies and testing positive for any allergies were associated with both asthma prevalence and asthma attack. b

Table 6 (Figure 2) presents the results of the multivariate analysis that included rhinitis symptoms, positive allergy testing, the interaction between rhinitis symptoms and positive allergy testing, and the odds of asthma, asthma attack, and wheezing. Rhinitis symptoms were associated with significant odds of asthma (OR: 4.79; CI 1.65–13.93; *p* < 0.01,). Children who had both a history of allergy and rhinitis symptoms had a slightly lower, but significant, odds of asthma (OR: 4.29; CI 1.08–17.03; *p* < 0.05). The odds of asthma attack among asthmatic children was insignificant for rhinitis symptoms or positive allergy test (at least one allergy). However, the odds of the interaction term between allergy and rhinitis symptoms was significant for asthma, suggesting asthmatic children with allergies and rhinitis were 15 times more like to experience asthma attack than those with both conditions (OR: 15.00; CI 1.00–224.9; *p* < 0.05).

## 4. Discussion

Three major findings emerged from this study: (a) allergies (especially environmental allergies), eczema, and allergic rhinitis showed strong associations with asthma prevalence and asthma severity; (b) the coexistence of multiple allergic comorbidities and multiple allergies were associated with increased risks of asthma and worsening of asthma symptoms, and (c) the average age of environmental allergy testing was significantly higher than that of asthma diagnosis. While the age difference in the asthma and allergy diagnosis makes a novel contribution, the overall findings of this study concerning the role of allergies in pediatric asthma are consistent with the literature. In addition, the findings of this paper have implications for understanding, mitigating, and managing asthma morbidity.

First, allergic asthma has been described as one of the most important phenotypes in childhood given its high prevalence, close association with early-onset of asthma that starts in childhood or early adolescence, and its tendency to persist into adulthood [2]. Our paper shows that multiple allergies are associated with asthma prevalence and asthma morbidity, consistent with the literature. Studies suggest that asthma is more common in patients with moderate to severe persistent rhinitis and that patients with severe/uncontrolled asthma have more severe nasal disease (often chronic rhinosinusitis) [12,13,14], and asthmatic patients with comorbid allergic rhinitis have increased medical resource utilization [12,13]. Moreover, eczema and allergic rhinitis are shown to play critical roles in the onset, severity, and persistence of pediatric asthma [5,15,16,17].

A causal link between allergic rhinitis and asthma is not yet established. However, some evidence suggests that both are on the same spectrum of the disease due to an underlying systemic allergic tendency [5]. Predictors of the progression of allergic rhinitis to asthma, along the “atopic march”, include repeated exposure to potent allergenic antigens, the most common of which is house dust mite [18,19,20]. In our study, allergies to animal dander, followed by mold and cockroach allergies, had the highest odds of asthma prevalence. However, these results should be interpreted with caution because our data were based on self-reporting of past allergy testing, as we did not assess the level of exposures to allergens over time. Given that there is a strong association between asthma and allergic conditions, it has clinical implications for the identification and management of both. To reduce morbidity and associated healthcare costs, attention is warranted to manage comorbid allergic conditions in children and potentially even early screening for allergies to manage them.

Second, other factors that play a role in the progression of allergic rhinitis to asthma include pets in the household, positive atopic family history, exposure to cigarette smoke in a dose-response fashion, female sex, and bronchial hyperreactivity [21,22]. Third, we observed that the average age for environmental allergy testing was 2 years higher than the average age of an asthma diagnosis and wheezing, despite the fact that allergies play a vital role in the onset of asthma, and its prevalence and severity. Few specific practical guidelines exist for pediatricians and family physicians on the indications for environmental allergy testing. The American Academy of Pediatrics recommends that testing only in the context of clinical history of allergies or in certain conditions, rather than a proactive preventive screening approach, as positive allergy testing may not always indicate a clinical allergy [23]. National asthma guidelines recommend that patients with persistent asthma be screened for allergies, with particular emphasis on testing for perennial indoor allergens (dust mite, cockroach, mold, pet dander) and outdoor allergens. These guidelines also suggest allergy testing in certain patients with asthma to increase education about allergen avoidance and possible immunotherapy [23]. Skin prick allergy tests may be performed in infants as young as 1 month, although children younger than 2 years are unlikely to be sensitized to as many allergens as older children, and positive reactions that do occur tend to be smaller and less reliable [24]. Our findings highlight the need for more comprehensive guidelines on the referral for allergy screening among young children, with particular attention to the timely referral of patients diagnosed with asthma.

Finally, improving (indoor) home environment is warranted to mitigate and manage both asthma morbidity and allergy symptoms. As evident from this paper, selected home environment factors were significantly associated with asthma, asthma severity and wheezing in our study. Recommended household measures for controlling allergen exposure include washing bedsheets weekly in hot water (>130 °F), using mattress and pillow allergen-impermeable covers, keeping pets out of the bedroom if removal of the pet from the home is unacceptable [25], maintaining an indoor temperature of 20 to 21 °C, and a relative humidity between 45 and 55%. Parents/guardians of children with asthma (diagnosis) are counseled about home control measures, such as washing sheets in hot water, regular change of air filter, and installing an air purifier. There are other likely latent household variables that warrant attention, such as exposure to first- and secondhand smoke. In this study, we did not find a significant association between smoke exposure and asthma morbidity and prevalence. But our sample included only 8 subjects who reported a smoker in their homes.

While this study makes novel contributions, there are limitations of this study. First, this study used surveys (though in clinical settings). Like any other survey-based study, it has inherent issues, such as recall bias, response error, and under- or over-reporting. Second, data used for this study were limited to parents of children attending two pediatric clinics, constraining the generalizability of our findings. The main outcome measures, namely asthma, wheezing, and allergy diagnoses, were reported by the participants. Therefore, the accuracy of these diagnoses cannot be verified. Third, only interested parents who had the time and adequate literacy level participated in the study, suggesting the plausibility of selection bias, though our response rate was more than 70%. Fourth, although the survey was offered in both English and Spanish, the diverse population of Miami means that certain clinic attendees were still left out, such as the significant Creole-speaking Haitian population. Fifth, our main outcome variable, i.e., asthma prevalence, may be subject to underreporting bias. For example, many children in our study, who did not have asthma diagnosed by a physician, reported wheezing during the last year. Sixth, due to the cross-sectional design, we could not assess the temporality of environmental exposures, sensitization to allergens, and the onset of allergic conditions with respect to asthma development. Finally, sample size was a limitation, particularly for analysis of asthma severity which was restricted to n = 36 for asthmatic children.

## 5. Conclusions

While overall findings of this paper are consistent with the literature, this paper advances the literature by demonstrating that multiple comorbidities and/or allergies coupled with home environment can increase the risk of asthma, its persistence, and severity, which has clinical implications for the management of asthma, allergies and associated allergic comorbidities. Because allergies and allergic comorbidities play a vital role in the onset and persistence of asthma, the importance of identifying and managing allergies in a timely manner, and medicinal therapies should be continually emphasized. Further understanding of parents’ awareness of household measures for allergy control is warranted, so that intervention for increased awareness and emphasis in clinical practice can be initiated as needed. Despite this contribution, further research is needed to clarify the mediational effects of home environment mediated allergies and allergic conditions. Because the home environment data in this study provided insight into the sources of environmental exposure, direct measurements of indoor air pollution and outdoor bioaerosols in the forms of pollens, mold spores, and other microorganisms are warranted in the future to quantify their precise effects on asthma severity and asthma persistence.

## Figures and Tables

**Figure 1 ijerph-18-04142-f001:**
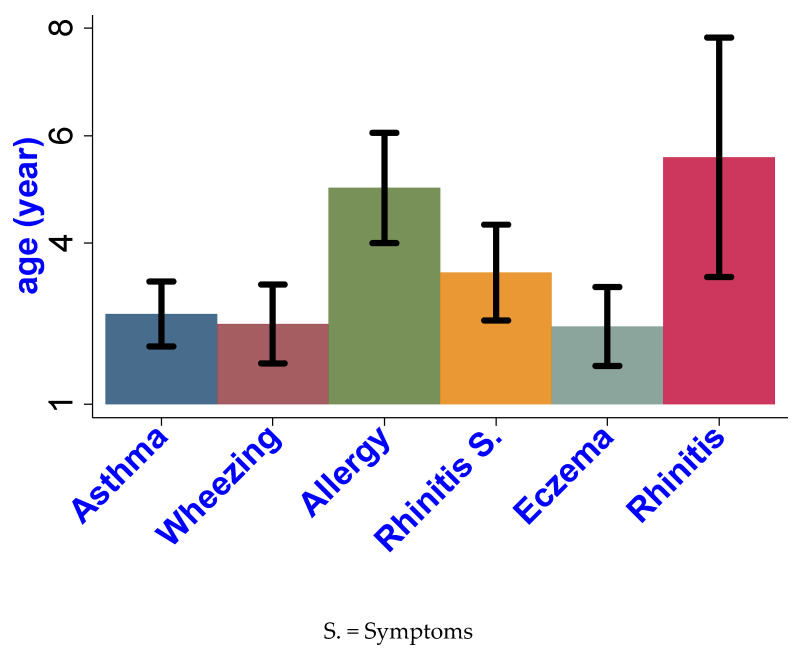
Mean age at diagnosis of allergic condition or onset of symptoms. Error bars indicate SDs. Wheezing is within the past year among children without asthma.

**Figure 2 ijerph-18-04142-f002:**
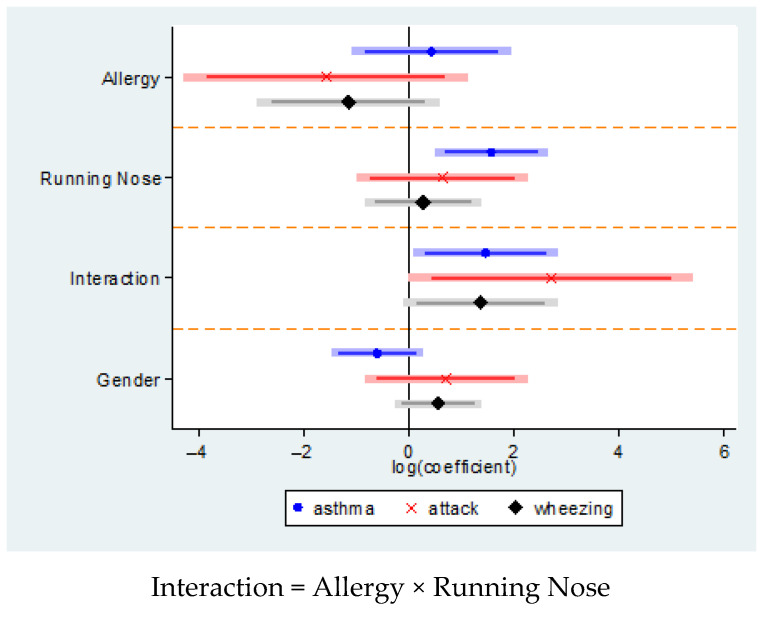
Risk of asthma, asthma attack, and wheezing (without asthma diagnosis) with respect to selected covariates.

**Table 1 ijerph-18-04142-t001:** Demographic and clinical characteristics of children attending two pediatrics clinics in Miami.

Variable Name	Number of Patients (N = 163)	% ^a^
Female	84	51.5
Age (Mean ± SD)	7.85 ± 4.84	-
0–3 years	32	19.9
4–7 years	42	26.1
8–11 years	50	31.1
12-18 years	37	23.0
Migrant to South Florida ^b^	18	11.0
Diagnosed asthma	36	22.1
Allergic rhinitis	15	9.6
Eczema	42	26.6
Rhinitis symptoms	43	25.7
Wheezing ever (among children without asthma) ^c^	39	30.7
Wheezing in past year ^c^	12	9.4
Tested for environmental allergies	35	22.9
Tested positive for any environmental allergen	18	11.0
Cat or dog dander ^d^	8	22.9
Mold ^d^	5	14.3
Dust mite ^d^	11	31.4
Cockroach ^d^	5	14.3
Pollen ^d^	13	37.1
Pets in household	79	48.5
Smoking in household	8	4.9
Home construction year		
<1970	35	25.9
1970–1990	47	34.8
1991- present	53	39.3
Child’s bedroom on 1st floor	103	73.6
Child’s bedroom on 2nd floor or higher	37	26.4
Temperature in home (Fahrenheit)		
60–70	19	12.7
71–75	96	64.0
76+	35	23.3
Temperature at which sheets are washed		
Cold	50	32.3
Warm	53	34.2
Hot	35	22.6
Air cleaner in home	49	33.1
Seen cockroach in home (past month)	66	42.9
Seen or smelled mold in home (past month)	8	4.8
Carpet in child’s bedroom	45	28.5
Parents pay attention to outdoor pollen at least sometimes	66	42.0
Parents accessed online pollen or allergy info in past week	20	14.8

^a^ Data are reported as percentage of all patients, unless otherwise indicated. ^b^ Was not born and raised in South Florida, which includes Miami. ^c^ Percentage is among children without asthma (*n* = 127). ^d^ Percentage is among children who were tested for environmental allergies (*n* = 35).

**Table 2 ijerph-18-04142-t002:** Mean age (year) of diagnosis of asthma and allergic comorbidities (95% confidence interval; # children in parenthesis).

Symptoms	Odds Ratio (95% CI; n)
Asthma	2.7
	(2.1 to 3.3; 36)
Wheezing	2.5
	(1.8 to 3.2; 39)
Allergies (including environmental)	5.0
	(4.0 to 6.1; 35)
Rhinitis Symptoms	3.5
	(2.6 to 4.3; 53)
Eczema	2.5
	(1.7 to 3.2; 42)
Allergic Rhinitis	5.6
	(3.37 to 7.8; 15)

**Table 3 ijerph-18-04142-t003:** Odds Ratio (95% confidence interval) of asthma prevalence, asthma attack and wheezing with respect to positive testing for different environmental allergens.

Covariate	Asthma	Wheezing ^b^	Asthma Attack
Mold	15.00 (1.5–150.2) ^a^	0.79 (0.1–7.4)	6.60 (0.5–82.3)
Dust mite	4.64 (1.3–16.8) ^a^	1.22 (0.3–4.9)	4.67 (0.6–33.7)
Cat or Dog dander	28.97 (2.9–286.3) ^a^	NE	1.43 (0.3–7.9)
Cockroach	15.00 (1.5–150.2) ^a^	0.79 (0.1–7.4)	6.60 (0.5–82.3)
Tree or grass pollen	9.75 (2.6–36.8) ^a^	0.56 (0.1–2.7)	1.60 (0.3–7.7)
Any positive allergen test (≥1 allergen)	6.39 (1.3–31.8) ^a^	0.47 (0.1–4.1)	0.67 (0.1–7.9)
Multiple positive allergens (≥2 allergens)	12.78 (2.9–57.2) ^a^	0.63 (0.1–3.1)	2.00 (0.4–10.6)

**^a^***p* <0.01. ^b^ Among children without asthma, within the past year.

**Table 4 ijerph-18-04142-t004:** Age and gender-adjusted odds of asthma, asthma attack, and wheezing with respect to selected covariates (95% confidence interval).

Covariate	Asthma Diagnosis	Wheezing ^a^	Asthma Attack ^a^	Asthma Severity Score ≥ 2 (n = 17) ^e^
Age	1.03 (0.94–1.13)	0.80 (0.63–1.01)	0.89 (0.73–1.08)	0.73 (0.58–0.91) **
Allergic comorbidities				
Allergic Rhinitis	10.13 (4.17–24.58) **	2.011 (0.91–4.44)	1.34 (0.31–5.87	1.09 (0.13–9.50)
Eczema diagnosis	2.67 (1.17–6.11) *	1.546 (0.67–3.60)	1.98 (0.47–8.34)	0.17 (0.03–1.07)
Rhinitis symptoms	6.39 (2.37–17.23) **	1.65 (0.61–4.48)	Not estimated ^d^	1.9 (0.26–14.44)
Use of anti-allergy medication (past month)	6.83 (2.84–16.47) **	0.91 (0.35–2.35)	0.98 (0.24–4.09)	1.50 (0.30–7.56)
Positive for ≥ 1 allergen	5.62 (2.02–15.69) **	0.99 (0.29–3.39)	1.75 (0.35–8.63)	0.40 (0.07–2.24)
Has been tested for environmental allergies	4.36 (1.82–10.47) **	1.67 (0.67–4.16)	2.93 (0.62–13.86)	
**Home Environment and Behaviors**				
Smoker in home	2.81 (0.64–12.32)	1.31 (0.24–7.10)	0.64 (0.07–5.73)	0.32 (0.02–4.15)
Pets in home	1.59 (0.71–3.57)	1.23 (0.58–2.59)	0.48 (0.11–2.06)	0.07 (0.01–0.58) *
Temperature at which sheets are washed (reference: cold)				
warm	1.51 (0.52–4.38)	3.32 (0.18–62.21)	0.50 (0.05–5.25)	0.02 (0.00–0.54) *
hot	3.32 (1.13–9.72) *	2.13 (0.13–33.91)	4.38 (0.51–37.51)	0.02 (0.00–0.68) *
Home temperature 76+ degrees F (reference: 60-70 F)	1.54 (0.40–5.93)	0.95 (0.82–1.10)	2.99 (0.21–41.99)	0.42 (0.02–9.81)
Parents pay attention to outdoor pollen at least sometimes	1.07 (0.49–2.34)	4.61 (0.60–35.32)	0.68 (0.16–2.89)	0.58 (0.11–2.94)
Parents accessed online pollen or allergy info in the past week	0.65 (0.19–2.23)	0.95 (0.28–3.24)	Not estimated	0.59 (0.05–6.95)

**^a^** Within the past year. Wheezing is among children without asthma diagnosis; ^d^ Odds ratio for rhinitis symptoms and asthma attack could not be estimated, but the relationship was but significant by chi-square association; 13/13 (100%) of patients that had an asthma attack in the past year reported rhinitis symptoms, compared with 13/21 (61.9%) that did not have an asthma attack, *p* < 0.05; ^e^ At least 2 of the 4 asthma severity indicators (asthma attack in past year, asthma symptoms in the past month, missed school in the past year, used asthma medication use in the past month); ^*^
*p <0.05;*
^**^
*p* <0.01

**Table 5 ijerph-18-04142-t005:** Factor loadings with main outcome variables.

Main Variables	Asthma Prevalence (*n* = 163)			Asthma Attack within Last Year (*n* = 36)		
	Factor1	Factor2	Factor3	Factor1	Factor2	Factor3
Outcome (1 = yes, 0 = No)	0.61	0.10	0.05	0.37	0.26	−0.05
Rhinitis (1 = yes, 0 = No)	0.68	0.37	0.09	0.45	0.70	−0.14
Running Nose (1 = yes, 0 = No)	0.47	0.49	−0.05	0.29	0.59	0.32
Eczema (1 = yes, 0 = No)	0.46	−0.12	−0.23	0.53	−0.17	0.06
Use of anti-histamine (1 = yes, 0 = No)	0.45	−0.09	0.03	0.23	−0.19	−0.06
Tested positive for allergy (1 = yes, 0 = No)	0.74	−0.32	0.14	0.70	−0.11	−0.50
Environmental Allergy (1 = yes, 0 = No)	0.72	−0.32	0.14	0.75	−0.18	−0.44
Use of pollen data (1 = yes, 0 = No)	−0.06	0.13	0.19	−0.31	0.46	−0.07
Paint in home last year (1 = yes, 0 = No)	0.14	−0.01	−0.19	0.43	0.29	0.25
Smoker in home (1 = yes, 0 = No)	0.23	0.29	−0.10	0.38	0.24	0.45
Pets in home (1 = yes, 0 = No)	−0.14	−0.15	−0.21	0.15	−0.22	0.55
Hay fever (1 = yes, 0 = No)	0.44	−0.04	−0.26	0.54	−0.29	0.31
Air filter change (Yes = 1, 0 otherwise)	−0.05	−0.02	0.43	−0.21	0.38	−0.56
Home temperature (1 = >72 °F, 0 otherwise)	−0.14	0.29	0.05	−0.10	0.45	0.15

**Table 6 ijerph-18-04142-t006:** Multivariate analysis. Age and gender-adjusted odds ratios (95% confidence interval) of asthma prevalence, asthma attack, and wheezing with respect to comorbid allergic conditions.

Covariate	Asthma	Asthma Attack	Wheezing ^a^
Tested positive for ≥ 1 allergen	1.53 (0.34–6.95)	0.21 (0.01–3.06)	0.31 (0.05–1.80)
Rhinitis symptoms	4.79 ** (1.65–13.93)	1.88 (0.36–9.73)	1.31 (0.43–3.97)
Allergy x rhinitis symptoms ^e^	4.29 * (1.08–17.03)	15.00 * (1.00–224.9)	3.91 (0.90–16.93)
Age category (1,5,10,15 y)	1.07 (0.97–1.18)	0.93 (0.76–1.14)	0.96 (0.88–1.05)
Female gender	0.55 (0.23–1.32)	2.02 (0.43–9.59)	1.74 (0.76–3.94)
# of observations	151	36	146

^a^ Among children without asthma, within the past year; * *p* < 0.05; ** *p* < 0.01. ^e^ Interaction term for allergy (history of positive test for at least 1 allergen) and rhinitis symptoms.

## Data Availability

The data used in this research are confidential and cannot be shared outside the study team. However, deidentified may be made available upon a reasonable request.

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
