# Peer review of "Allergies, Allergic Comorbidities and the Home Environment in Pediatric Asthma in Southern Florida"

_ijerph, 2021, doi:10.3390/ijerph18084142_

Round 1

Reviewer 1 Report

Saif et al. present their findings of a prospective, cross-sectional survey assessing for atopic conditions and known environmental associations. The authors have done excellent job of concisely and clearly presenting the results of this study. While the study findings largely involves known associations, the authors factor analysis gives a nice overview of the correlation between atopic and environmental conditions which might be associated with asthma.

I have two minor comments, but otherwise this is well written article.

1) Did the authors collect any information about about race/ethnicity? This might be interesting as I might imagine the larger percentage of the population that might Hispanic/Latinx/Caribbean which are often under represented in other cohorts investigating asthma/allergies

2) The average age of diagnosis of asthma is was below the age at which spirometry can be performed and many individuals might only have transient symptomology. I might be interesting perform the factor analysis stratified by age of onset if there are sufficient numbers.

Author Response

Our responses to your comments begin with "AR:"

Did the authors collect any information about race/ethnicity? This might be interesting as I might imagine the larger percentage of the population that might Hispanic/Latinx/Caribbean which are often under represented in other cohorts investigating asthma/allergies

AR: Thank you for this helpful comment – certainly a crucial factor to consider, unfortunately we did not collect race/ethnicity data for this study but should certainly be included in future studies especially given the demographics of and health disparities in South Florida.

2) The average age of diagnosis of asthma was below the age at which spirometry can be performed and many individuals might only have transient symptomology. It might be interesting to perform the factor analysis stratified by age of onset if there are sufficient numbers.

AR: We did not administer spirometry. Asthma diagnosis and age of diagnosis was confirmed by children’s parents/guardians.

Reviewer 2 Report

An interesting paper which addresses the relation between asthma, rhinitis and concomitant allergies. This kind of studies was done before in other populations, but it is always interesting and relevant to be aware of what allergens are most relevant in your area of practice, as differences between geographical regions can be very high. 

Author Response

Our responses begin with "AR:"

An interesting paper which addresses the relation between asthma, rhinitis and concomitant allergies. This kind of studies was done before in other populations, but it is always interesting and relevant to be aware of what allergens are most relevant in your area of practice, as differences between geographical regions can be very high.

AR: Thank you for recognizing regional contribution of our paper.

Reviewer 3 Report

General comments

This study is based on retrospective questionnaire answered in the waiting-room of a doctor’s office, instead of using a structured history and diagnose asthma, atopic asthma allergic rhino-conjunctivitis etc. by the doctor in that office (?). The better defined diseases the more true information will be obtained.

If you will continue with similar studies, my advice is to use better defined diseases. Asthma has multiple causes and should be subdivided into subgroups e.g. as you mention atopic asthma that is the most common form among schoolchildren, the relatively few children without allergy but with severe asthma and wheeze in infants without allergies disappearing before school age, etc., but they must be well defined. .

Specific Comments

Line                Comment

49                    Should  be ” … with doctors diagnosed asthma …”

50                    Should be ” …  without known allergies …

49-50              Those with asthma attack had asthma, even if not atopic asthma with thecriteria set up in your study. IgE tests from the major supplier are mostly well standardized. However, skin tests rely on allergen extract potency, composition, technician etc..

68-9                Immunologically mediated hypersensitivity is called allergy. Allery mediated by IgE is named IgE mediated allergy and allergy mediated after very low dose exposure to allergens atopic allergy (1, 2).

70                    The percentage depends e.g. at the dose of exposure, the potency and composition of the allergen extract used for definition of “atopic allergy”, the skill of the testing technician etc.

78                    The question is if allergic rhinitis is a comorbidity to atopic asthma or the opposite. In patients with atopic asthma, sensitization causes positive challenge tests in both the nose and the conjunctiva in the majority of cases.

90                    Only an epidemiologist can call a study based on retrospective questionnaires for prospective. This is a major flaw causing misclassification of many patients. See above.

105 -  ff          Using retrospective, non-defined criteria for diagnosis is not optimal for this type of studies.         

131-2              The selection of the patient sample investigated is not sufficiently described. Did the doctor in that office see patients with suspected allergies or was he/she a pediatrician or even GP? Did the children attending the office have common cold? CF? neurological disorders or were all of them allergic or with suspected allergies?

Table 1           You must inform on how many did not respond to each question. E.g. the Temperature at home was responded by 50 not 163. similarly, the temp of the washing machine was responded by 138, seen cockroach 154, Air cleaner 148 the child’s bedroom 140 etc. You use a column to the right for percentages. Despite that you add %even in the other column for 6 items (?).

Figure 1          Here and in most of the manuscript you use allergy instead of sensitization. A test does not mean clinical allergy! Nowhere, you explain what U. Rhinitis means aand how you differ between this “diagnosis” and Rhinitis, Does thinitis mean common cold??

1b                    In m opinion Table 2. It is connected with Figure 2, I think. I tink the tables must be re-numbered.

Table 2           I highly question this table based on the mothers memory of the information from another doctor . I do not believe the mothers remember The names of all allergens mentioned, especially not the molds. Sensitization must rely on actual tests performed according to strict protocols with known agents etc.

Figure 2          The difference between asthma and asthma attack is only the doctors diagnosis. Or?

References

  1. Johansson SGO, Bieber T, Dahl R, Friedmann PS, Lanier B, Q., Lockey RF, et al. Revised nomenclature for allergy for global use: Report of the Nomenclature Review Committee of the World Allergy Organization, October 2003. J Allergy Clin Immunol. 2004;113(5):832-6.
  2. Johansson SGO, Hourihane JO, Bousquet J, Bruijnzeel-Koomen C, Dreborg S, Haahtela T, et al. A revised nomenclature for allergy. An EAACI position statement from the EAACI nomenclature task force. Allergy. 2001;56(9):813-24.

Author Response

Your comments are pasted as is, and our responses begin with "AR:"

This study is based on retrospective questionnaire answered in the waiting-room of a doctor’s office, instead of using a structured history and diagnose asthma, atopic asthma allergic rhino-conjunctivitis etc. by the doctor in that office (?). The better defined diseases the more true information will be obtained.

AR: Yes this is certainly a limitation of a self-reported survey based research, and a chart review could be considered in the future to address this issue. However, it was beyond the scope of this paper.

If you will continue with similar studies, my advice is to use better defined diseases. Asthma has multiple causes and should be subdivided into subgroups e.g. as you mention atopic asthma that is the most common form among schoolchildren, the relatively few children without allergy but with severe asthma and wheeze in infants without allergies disappearing before school age, etc., but they must be well defined. .

AR: You are correct, many children do outgrow their asthma (and potentially allergies) with age. This will be important consideration to ask children, especially older children, when their asthma and allergy symptoms stopped.

Specific Comments

Line                Comment

49                    Should  be ” … with doctors diagnosed asthma …”

AR: Edited as suggested.

50                    Should be ” …  without known allergies …

AR: Edited as suggested.

49-50    Those with asthma attack had asthma, even if not atopic asthma with the criteria set up in your study. IgE tests from the major supplier are mostly well standardized. However, skin tests rely on allergen extract potency, composition, technician etc..

In this paper, allergies diagnosis was reported by children’s parents/guardians based on responses to two questions: “Has a doctor or other health professional ever told your child that he/she has allergies?” and “Has your child been tested for “environmental allergies?”. If responses were yes, then we asked follow up questions concerning the allergy types. We acknowledge that it can be subject to reporting bias, and we have mentioned it in the limitations (see lines 318-336).

68-9                Immunologically mediated hypersensitivity is called allergy. Allery mediated by IgE is named IgE mediated allergy and allergy mediated after very low dose exposure to allergens atopic allergy (1, 2).

AR: Given the data constraint it was not feasible to make such a distinction. However, it was used in the context of defining atopic and non-atopic asthma. To make it clear we removed allergic to “atopic asthma induced by external triggers, including environmental allergens” (see line 70).

70                    The percentage depends e.g. at the dose of exposure, the potency and composition of the allergen extract used for definition of “atopic allergy”, the skill of the testing technician etc.

AR: Given we did not run skin tests, it was not feasible to identify and assess the limitations of self-reported allergy status of children included in the study.

78                    The question is if allergic rhinitis is a comorbidity to atopic asthma or the opposite. In patients with atopic asthma, sensitization causes positive challenge tests in both the nose and the conjunctiva in the majority of cases.

AR: We agree. However, given the nature data of our study there was no way confirm this. But did include allergic rhinitis within allergic comorbidities.

90                    Only an epidemiologist can call a study based on retrospective questionnaires for prospective. This is a major flaw causing misclassification of many patients. See above.

AR: Revised to “cross-sectional study” from prospective study

105 -          Using retrospective, non-defined criteria for diagnosis is not optimal for this type of studies.

 AR: understood! 

131-2              The selection of the patient sample investigated is not sufficiently described. Did the doctor in that office see patients with suspected allergies or was he/she a pediatrician or even GP? Did the children attending the office have common cold? CF? neurological disorders or were all of them allergic or with suspected allergies?

AR: Thank you for this comment – data were collected at two general pediatrics clinics; lines 94-97 were revised to make this more clear.

Table 1           You must inform on how many did not respond to each question. E.g. the Temperature at home was responded by 50 not 163. similarly, the temp of the washing machine was responded by 138, seen cockroach 154, Air cleaner 148 the child’s bedroom 140 etc. You use a column to the right for percentages. Despite that you add %even in the other column for 6 items (?).

AR: Deleted redundant % in the column for those 6 items

Figure 1          Here and in most of the manuscript you use allergy instead of sensitization. A test does not mean clinical allergy! Nowhere, you explain what U. Rhinitis means and how you differ between this “diagnosis” and Rhinitis, Does rhinitis mean common cold??

AR: Rhinitis was meant to indicate symptoms of non-infectious rhinitis (the survey question was asking if a child has “had a problem with sneezing, or a runny, blocked nose when he/she did not have the cold or the flu”). This has been clarified in the manuscript

AR:

1b                    In my opinion Table 2. It is connected with Figure 2, I think. I think the tables must be re-numbered.

AR: Table 2 is not connected to Figure 2. Because Table two shows the odds ratio of univariate results, and Figure 2 shows the results of multivariate analysis.

Table 2           I highly question this table based on the mothers memory of the information from another doctor . I do not believe the mothers remember The names of all allergens mentioned, especially not the molds. Sensitization must rely on actual tests performed according to strict protocols with known agents etc.

AR: Certainly true – a comment was added about this limitation in the discussion (see lines 318-336).

Figure 2          The difference between asthma and asthma attack is only the doctors diagnosis. Or?

AR: Yes – the asthma attack questions (ER or urgent care visit in past year for asthma, parent’s report of asthma attack in the past year) was asked of parents who reported their child had physician-diagnosed asthma

Reviewer 4 Report

This study described the association of allergic comorbidities and the home environment to pediatric asthma from a cross-sectional study recruited in Florida. The rational and study design is adequate. However, there are several flaws in the method description and analysis strategies that needs to be addressed.

  1. The description of the aim of the study in the introduction and the “Objective” in the abstract seems quite laboured and should be more concise.  
  2. Line 75/76: What are the “associated comorbidities”? Likewise, consider adapting the title from “Allergies and Associated comorbidities…” to “Allergic comorbidities …” since no other comorbidities were investigated in this study.
  3. The last sentence of the introduction (Line 84-87) does not to fit here.
  4. The question how wheezing was assessed should be included in the methods.
  5. A much more detailed description of the applied statistics in the methods should be included. Can you please explain why the multivariate model was used? Please define the control groups used in each analysis, e.g. are the asthmatics compared to no asthmatics (including wheezing children), or to no wheeze/no asthma? Also, provide case numbers for each groups you are comparing. This information should also be provided in each table.
  6. Table 1b: It is not clear which statistical test was used. Include p-value for the comparison. Adding more horizontal lines in the table would facilitate reading.
  7. Figure 1: Include p-values and applied statistical tests in the figure. It is not clear which test was used in Line 158. The presentation of the same results two times in Figure 1 and Table 1b might not be necessary - or was a different analysis method applied with subsequent different findings? If so, the authors should better describe these different results.
  8. There a number of elements of the analyses that are unclear, or require further description. I am not familiar with this type of analysis used in Table 4, and I assume other readers may not be either. I’m not sure whether the presentation is adequate. Again, a more detailed description of the statistics would help here.
  9. Line 161-163 and line 233: This statement might be misleading. Are the children tested positive for one allergy only, or >= one allergy?
  10. What is the difference between the analysis of Table 5 and Figure 2?  
  11. Indicate the case numbers (for each group) as well as the applied statistical model/test in each tables and figure.
  12. What is the difference between hay fever and rhinitis in Table 4 or between “tested pos. for allergy” and “environmental allergy”, respectively?
  13. Do the authors have any explanation why smoking at home was not significantly associated with asthma, wheezing or asthma attack?
  14. The discussion appears verbose. A clear structure guiding the reader through the discussion is missing. What are the author’s main messages?
  15. Line 315: Why are the two different clinics not included as an additional confounder/covariate in the model to account for this potential bias? Was there any difference between the two recruitment centers?

Minor:

  1. Figure 2 is not mentioned in the text.
  2. What means “**” in table 5?
  3. Please, indicate the study type in the abstract.
  4. Line 74: Could reference 4 be substituted by a more recent one?
  5. I recommend changing the general word “comorbidity” by “allergic comorbidities” throughout the text.
  6. Since the publication is intended for an international readership, the use of °C instead of °F is recommended.
  7. Line 131 and 136: Consider replacing “study sample” by “study participant”
  8. Table 1a: Unit for temperature is missing.
  9. Numbers ranging from 1 to 10 should be spelled out, e.g.Line 161, heading table 1a.
  10. Line 166-177: The text paragraph should refer to table 3 including the corresponding results.
  11. Figure 1: label of y-axis is missing. What means “Allergy” here?
  12. Table 3: The “a” appears two times with different meaning. “b” or “c” is missing for the OR of “hot washed”-asthma association.

Author Response

Your comments are pasted as is, our responses begin with "AR:".

This study described the association of allergic comorbidities and the home environment to pediatric asthma from a cross-sectional study recruited in Florida. The rational and study design is adequate. However, there are several flaws in the method description and analysis strategies that needs to be addressed.

The description of the aim of the study in the introduction and the “Objective” in the abstract seems quite laboured and should be more concise. 

AR: Revisions made to address this concern.

Line 75/76: What are the “associated comorbidities”? Likewise, consider adapting the title from “Allergies and Associated comorbidities…” to “Allergic comorbidities …” since no other comorbidities were investigated in this study.

AR: Revision made to “allergic comorbidities”.

The last sentence of the introduction (Line 84-87) does not to fit here.

AR: Edited to a different section of the introduction

The question how wheezing was assessed should be included in the methods.

AR: Edit made

A much more detailed description of the applied statistics in the methods should be included. Can you please explain why the multivariate model was used? Please define the control groups used in each analysis, e.g. are the asthmatics compared to no asthmatics (including wheezing children), or to no wheeze/no asthma? Also, provide case numbers for each groups you are comparing. This information should also be provided in each table.

AR: We expanded this section (see line 129-138). The multivariate analyses were critical to avoid any confounding due to age, gender and other related variables. You are correct that we compared asthmatic to non-asthmatic, and non-asthmatic included children with wheezing. I know this is a major limitation and we have highlighted this limitation in the discussion (see line 318-336). 

Table 1b: It is not clear which statistical test was used. Include p-value for the comparison. Adding more horizontal lines in the table would facilitate reading.

AR: These are mean ages (no p values); added additional horizontal lines to the table, and we did not ran any comparison in the age of diagnosis of different outcomes:

Figure 1: Include p-values and applied statistical tests in the figure. It is not clear which test was used in Line 158. The presentation of the same results two times in Figure 1 and Table 1b might not be necessary - or was a different analysis method applied with subsequent different findings? If so, the authors should better describe these different results.

AR: Correct, both figure 1 and table 1b are displaying the same information. However, Table 1b also provides exact confidence interval and sample of each condition).

There a number of elements of the analyses that are unclear, or require further description. I am not familiar with this type of analysis used in Table 4, and I assume other readers may not be either. I’m not sure whether the presentation is adequate. Again, a more detailed description of the statistics would help here.

AR: This method is detailed in the Methods and Material section (see lines 130-134)

Line 161-163 and line 233: This statement might be misleading. Are the children tested positive for one allergy only, or >= one allergy?

AR: Edit made – at least one allergy

What is the difference between the analysis of Table 5 and Figure 2? 

AR: Virtually these is no difference in between these two. While Table provides more details, and Figure 2, shear with a visual inspection, provides an overall trends of the results.

Indicate the case numbers (for each group) as well as the applied statistical model/test in each tables and figure.

AR: Number of observation is included in the list line of Table 5.

What is the difference between hay fever and rhinitis in Table 4 or between “tested pos. for allergy” and “environmental allergy”, respectively?

AR:  Please see bellow

Hay fever = parent’s report of physician diagnosed

Rhinitis = Rhinitis symptoms

Tested positive for allergy = parent’s report of positive to at least 1 allergen on testing in the past

Environmental allergy = parent’s report of physician diagnosis of environmental allergens

Do the authors have any explanation why smoking at home was not significantly associated with asthma, wheezing or asthma attack?

AR: Thank you for this comment – a sentence was added to the discussion section to address this (small n for smokers in household and potential stigma leading to underreporting)

The discussion appears verbose. A clear structure guiding the reader through the discussion is missing. What are the author’s main messages?

AR: We have rewritten discussion to improve readability and provided a structure to it: summary of the main findings, their implication and the study limitations, and separated the conclusion from the discussion.

Line 315: Why are the two different clinics not included as an additional confounder/covariate in the model to account for this potential bias? Was there any difference between the two recruitment centers?

AR: The authors felt the populations served by each clinic were similar enough to not warrant mention of this as a major confounder, as both were general pediatric clinics under the University of Miami system in downtown Miami (in neighborhoods of similar demographic makeup)

Minor:

Figure 2 is not mentioned in the text.

What means “**” in table 5?

AR: ** are defined!

Please, indicate the study type in the abstract.

AR: Edit completed

Line 74: Could reference 4 be substituted by a more recent one?

AR: Unfortunately there is not great more recent worldwide data highlighting these trends among children (ISAAC phase 3 study), but more recent data on asthma should be forthcoming via the Global Asthma Network Phase I

I recommend changing the general word “comorbidity” by “allergic comorbidities” throughout the text.

AR: Edit completed

Since the publication is intended for an international readership, the use of °C instead of °F is recommended.

AR: Unfortunately given cumbersome decimal points in switching over, opted to keep Fahrenheit

Line 131 and 136: Consider replacing “study sample” by “study participant”

AR: Edit completed

Table 1a: Unit for temperature is missing.

AR: Edited - Fahrenheit

Numbers ranging from 1 to 10 should be spelled out, e.g. Line 161, heading table 1a.

AR: edited as suggested.

Line 166-177: The text paragraph should refer to table 3 including the corresponding results.

Figure 1: label of y-axis is missing. What means “Allergy” here?

AR: It looks like, system created PDF skips exporting y-axis. Please download word version of our manuscript and it should show up. Allergy, means, a physician diagnosed allergy reported by children’s parents/guardians.

Table 3: The “a” appears two times with different meaning. “b” or “c” is missing for the OR of “hot washed”-asthma association.

AR: Edit completed

Reviewer 5 Report

Thank you for the opportunity to review the presented paper.

I found it very clear with a thorough description of all methodological issues.

An examined sample is rather moderate in size however sufficient for planned analyses  - a total of 163 parents/guardians completed the survey.

Questions were based on well-established questionnaire tools.

Outcomes of interest included asthma diagnosis, undiagnosed asthma, and asthma severity.

Risks of asthma, asthma severity indicators, and wheezing in the past year among non-asthmatic children (potentially undiagnosed asthma) were examined with respect to allergic comorbidities, home environment, and environmental allergens using multivariate logistic regression. All risk estimates were adjusted for age and gender.

Risk analyses included age and gender as basic confounders.

Minor questions

Authors categorized the study as “A prospective cross-sectional design”. Could you explain the reasons? For me, it is just a “cross-sectional design”.

Sample of children examined – please explain why the sample was limited to  91 consenting parents of children attending two general pediatric clinics at the University?  Is there any risk of selection bias

Author Response

Risk analyses included age and gender as basic confounders.

AR: Thank you for reading this manuscript and your comments.

Minor questions

Authors categorized the study as “A prospective cross-sectional design”. Could you explain the reasons? For me, it is just a “cross-sectional design”.

AR: Edited to “cross-sectional” design

Sample of children examined – please explain why the sample was limited to  91 consenting parents of children attending two general pediatric clinics at the University?  Is there any risk of selection bias

AR: Yes there is risk of selection bias which we acknowledged in our discussion, as only those parents in the waiting room who had the time and consented to participate in the study were included (see lines 318-336).

Round 2

Reviewer 3 Report

0

Reviewer 4 Report

The authors substantially improved the manuscript. I have one minor comment.

Line 136: Remove one „ to identify”